# Few-shot Task-agnostic Neural Architecture Search for Distilling Large Language Models

**Dongkuan Xu**[*]
NC State University
dxu27@ncsu.edu

**Subhabrata Mukherjee**[†]
Microsoft Research
submukhe@microsoft.com

**Xiaodong Liu**
Microsoft Research

**Debadeepta Dey**
Microsoft Research

**Wenhui Wang**
Microsoft Research

**Xiang Zhang**
Penn State University

**Ahmed Hassan Awadallah**
Microsoft Research

**Jianfeng Gao**
Microsoft Research

## Abstract

Traditional knowledge distillation (KD) methods manually design student architectures to compress large models given pre-specified computational cost. This requires several trials to find viable students, and repeating the process with change in computational budget. We use Neural Architecture Search (NAS) to automatically distill several compressed students with variable cost from a large model. Existing NAS methods train a single SuperLM consisting of millions of subnetworks with weight-sharing, resulting in interference between subnetworks of different sizes. Additionally, many of these works are task-specific requiring task labels for SuperLM training. Our framework `AutoDistil` addresses above challenges with the following steps: (a) Incorporates inductive bias and heuristics to partition Transformer search space into $K$ compact sub-spaces (e.g., $K$=3 can generate typical student sizes of base, small and tiny); (b) Trains one SuperLM for each sub-space using task-agnostic objective (e.g., self-attention distillation) with weight-sharing of students; (c) Lightweight search for the optimal student without re-training. Task-agnostic training and search allow students to be reused for fine-tuning on any downstream task. Experiments on GLUE benchmark demonstrate `AutoDistil` to outperform state-of-the-art KD and NAS methods with upto 41x reduction in computational cost. Code and models are available at aka.ms/autodistil.

## 1 Introduction

While large pre-trained language models (e.g., BERT [1], GPT-3 [2]) are effective, their huge size poses significant challenges for downstream applications in terms of energy consumption and cost of inference [3] limiting their usage in on the edge scenarios and under constrained computational inference budgets. Knowledge distillation [4, 5, 6, 7] has shown strong results in compressing pre-trained language models into small student models. However, these works require pre-specification of the student architecture and computational cost (e.g., number of parameters, FLOPs) for distillation. This poses two significant challenges: (i) it requires several trials to come up with viable architectures as they are hand-engineered and to define several hyper-parameters (e.g., number of layers and attention heads, hidden dimension, etc.); (ii) one has to re-run distillation with any change in specification for the student architecture or computational cost for using it in a target environment.

---

[*]Work done while interning at Microsoft Research.
[†]Corresponding author and project lead.

36th Conference on Neural Information Processing Systems (NeurIPS 2022).

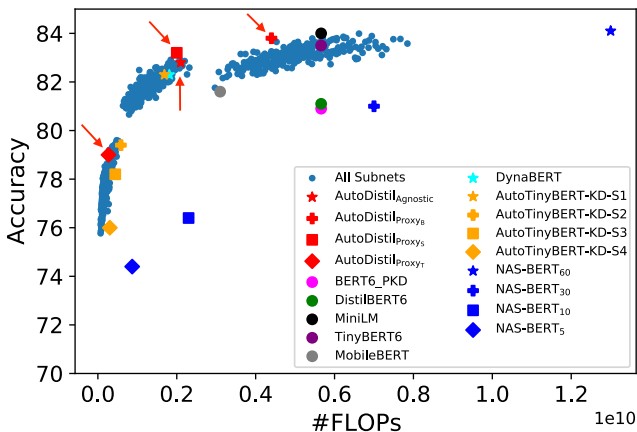

Figure 1: `AutoDistil` uses few-shot task-agnostic NAS to distill several compressed students with variable #FLOPs (x-axis) from $K$=3 SuperLMs (corresponding to each point cloud) trained on $K$ sub-spaces of Transformer search space. Each student (blue dot) extracted from the SuperLM is fine-tuned on MNLI with accuracy on y-axis. The best student from each SuperLM is marked in red. Given any state-of-the-art distilled model, `AutoDistil` generates a better candidate with less #FLOPs and improved task performance from corresponding search space. More discussions on trade-off and selection strategies can be found in the Appendix (Sections G, H).

Neural Architecture Search (NAS) [8, 9, 10, 11] provides a natural solution to automatically search through a large space of candidate models.The dominant NAS paradigm consists of two main steps: (a) Training a Super model combining all possible architectures into a single graph and jointly training them via weight-sharing; (b) Searching for optimal architecture from Super model with best accuracy on a downstream task, satisfying user-specified latency constraint for target device. Parallel to above computer vision (CV) works, NAS has shown strong results in recent works like DynaBERT [12], AutoTinyBERT [13] and NAS-BERT [14] for natural language understanding (NLU).

**Drawbacks of existing NAS methods.**
**[D1: Co-adaptation in weight-sharing]** Above works train one single large Super Language Model (SuperLM) consisting of millions of diverse student architectures. This results in some undesirable effects of co-adaptation [15] like conflicts in weight-sharing where bigger student models converge faster in contrast to smaller ones converging slower [16, 11].
**[D2: Multi-stage training]** A single SuperLM may not have sufficient capacity to encode a large search space. As a result, these works use multi-stage training process, where they first conduct NAS to identify candidate students and then perform further pre-training [13] and knowledge distillation [14] of the candidates.
**[D3: Task-specific training]** NAS works in CV domain (e.g., AutoFormer [17], Once-for-all [10], One-Shot NAS [11, 18]) leverage hard class labels from a task (e.g., image classification) or soft labels from ImageNet pre-trained models (e.g., MobileNet [7], RegNet [19]) for task-specific optimization with accuracy as evaluation metric. Different from CV domain, NLU tasks have different objectives and evaluation metrics for classification (e.g., MNLI), regression (e.g., STS-B) and correlation (e.g., CoLA). This makes it challenging to adapt existing NAS works to the NLU domain in a task-agnostic setting. Recent NAS works in the NLU domain are not fully task-agnostic. For instance, DynaBERT [12] accesses both task labels for knowledge distillation and task development set for network rewiring. NAS-BERT [14] performs two-stage knowledge distillation with pre-training and fine-tuning of the candidates. While AutoTinyBERT [13] also explores task-agnostic training, we demonstrate better performance from few-shot NAS and much cheaper cost from single stage training without additional pre-training and distillation. We present a detailed discussion on the differences between `AutoDistil` and existing KD and NAS methods based on fine-grained search space, different training strategies and amortized training cost in Table 1. More comparison details are summarized in Section A of Appendix.

**Contributions.** We address above challenges with fully task-agnostic few-shot NAS consisting of three steps. **(S1) Search space design.** We partition the Transformer search space into $K$ sub-spaces considering important architectural hyper-parameters like the network depth, width and attention heads. We further leverage inductive bias and heuristics to limit the number of student architectures in each sub-space. **(S2) Fully task-agnostic SuperLM training.** We train $K$ SuperLM overall, one

Table 1: Comparing `AutoDistil` with existing KD and NAS methods on aspects as task-agnostic training; generating students with variable compression cost; single-stage training without additional adaptation; SuperLM training with compact search space to mitigate interference ($P$ denotes partial).

| Method | Task-agnostic | Variable Compression | NAS | | |
|---|---|---|---|---|---|
| | | | Single Stage | SuperLM Training | Compact Search |
| BERT-PKD | ✗ | ✗ | | | |
| SparseBERT | ✗ | ✗ | | | |
| DistilBERT | ✓ | ✗ | | N/A | |
| TinyBERT | ✓ | ✗ | | | |
| MOBILEBERT | ✓ | ✗ | | | |
| MINILM | ✓ | ✗ | | | |
| DynaBERT | ✗ | ✓ | ✓ | One-shot | ✗ |
| NAS-BERT | $P$ | ✓ | ✗ | One-shot | ✗ |
| AutoTinyBERT | $P$ | ✓ | ✗ | One-shot | ✗ |
| `AutoDistil` | ✓ | ✓ | ✓ | Few-shot | ✓ |

for every sub-space. This allows each SuperLM more capacity to encode a sub-space as opposed to a single large one. We train each SuperLM with a fully task-agnostic objective (without accessing any task labels) like deep self-attention distillation, where we transfer knowledge from the self-attention module (including keys, queries and values) of a pre-trained teacher (e.g., BERT) to the student and use weight-sharing to train the SuperLM. **(S3) Lightweight optimal student search.** We obtain optimal student(s) directly from well-trained SuperLM(s) without any re-training that can be *simply fine-tuned* on downstream tasks. Our contributions over existing NAS works can be summarized as:

- In contrast to prior works (e.g., DynaBERT, AutoTinyBERT, NAS-BERT), we do a single-stage training combining NAS and distillation with no further pre-training or augmentation and demonstrate superior performance of the NAS process itself with significantly reduced training cost. Obtained subnetworks are simply fine-tuned on downstream tasks.

- Fully task-agnostic training with subnetwork attention state alignment for self-attention relation distillation and search in contrast to prior works in NLU (e.g., DynaBERT, NAS-BERT) and CV (e.g., AutoFormer, BigNAS, Once-For-All).

- Few-shot NAS to mitigate gradient conflicts in SuperNet training compared to prior One-shot NAS works in NLU (e.g., DynaBERT, AutoTinyBERT, NAS-BERT). AutoFormer in the CV domain is an exception to this point which also uses few-shot NAS but accesses task labels during training.

- Strong results over all the above NAS and distillation works in NLU with $3x$ additional compression over best performing distillation technique with negligible performance drop.

## 2   Background

We present an overview of Transformers [20], especially its two main sub-layers, multi-head self-attention (MHA) and feed-forward network (FFN). Transformer layers are stacked to encode contextual information for input tokens as: $\mathbf{X}^l = \text{Transformer}_l(\mathbf{X}^{l-1})$, $l \in [1, L]$ where $L$ is the number of Transformer layers, $\mathbf{X}^l \in \mathbb{R}^{s*d_{hid}}$, $s$ is the sentence length, and $d_{hid}$ is the hidden dimension. In the following, we omit the layer indices for simplicity.

**Multi-Head Self-Attention (MHA).** Given previous Transformer layer's output $\mathbf{X}$, MHA computes:

$$\text{Attention}(\mathbf{Q}_h, \mathbf{K}_h, \mathbf{V}_h) = \text{softmax}(\frac{\mathbf{Q}_h \mathbf{K}_h^\top}{\sqrt{d_{head}}})\mathbf{V}_h; \; \mathbf{Q}_h, \mathbf{K}_h, \mathbf{V}_h = \mathbf{X}\boldsymbol{W}_h^Q, \mathbf{X}\boldsymbol{W}_h^K, \mathbf{X}\boldsymbol{W}_h^V, \quad (1)$$

$$\text{MHA}(\mathbf{X}) = \text{Concat}(\text{head}_1, \cdots, \text{head}_H)\boldsymbol{W}^O, \quad (2)$$

where $\boldsymbol{W}_h^Q, \boldsymbol{W}_h^K, \boldsymbol{W}_h^V \in \mathbb{R}^{d_{hid}*d_{head}}$, $\boldsymbol{W}^O \in \mathbb{R}^{d_{hid}*d_{hid}}$ are linear transformations. $\mathbf{Q}_h, \mathbf{K}_h, \mathbf{V}_h \in \mathbb{R}^{s*d_{head}}$ are called queries, keys, and values, respectively. $H$ is the number of heads. $\text{head}_h = \text{Attention}(\mathbf{Q}_h, \mathbf{K}_h, \mathbf{V}_h)$ denotes the $h$-th attention head. $\text{Concat}$ is the concatenating operation. $d_{head} = d_{hid}/H$ is the dimension of each head.

**Feed-Forward Network (FFN).** Each Transformer layer contains an FNN sub-layer, which is stacked on the MHA. FFN consists of two linear transformations with a ReLU activation as:

$$\text{FFN}(x) = \max(0, x\boldsymbol{W}^1 + b_1)\boldsymbol{W}^2 + b_2, \quad (3)$$

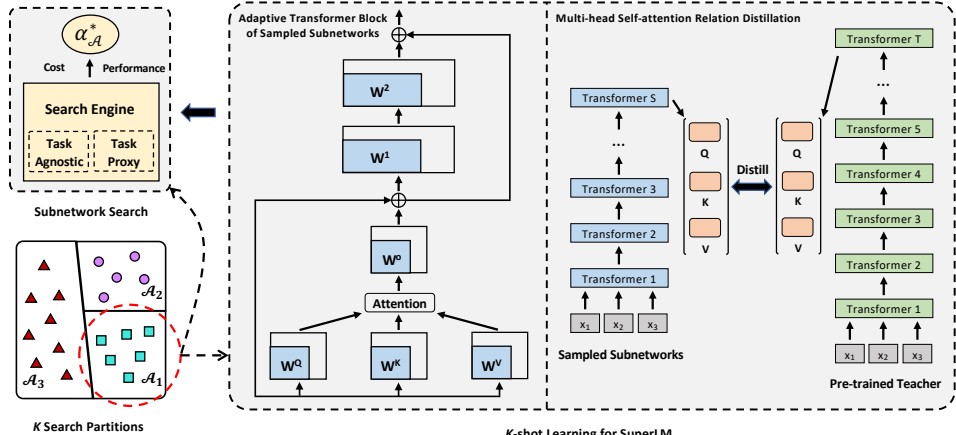

Figure 2: Overview of `AutoDistil`. It considers $K$ partitions of the Transformer architecture subspace to train one SuperLM for each partition with weight-sharing of the constituent subnetworks trained via task-agnostic deep self-attention distillation. Optimal compressed subnetworks can be easily extracted from the SuperLMs without additional training or distillation.

where $\boldsymbol{W}^1 \in \mathbb{R}^{d_{hid}*d_f}$, $\boldsymbol{W}^2 \in \mathbb{R}^{d_f*d_{hid}}$, $b_1 \in \mathbb{R}^{d_f}$, and $b_2 \in \mathbb{R}^{d_{hid}}$. In addition, there are residual connection and layer normalization on top of MHA and FFN (denoted by $\oplus$ in Figure 2), which are formulated as LayerNorm(x + MHA(x)) and LayerNorm(x + FFN(x)), respectively.

# 3 Few-shot Task-agnostic NAS

Given a large pre-trained language model (e.g., BERT) as teacher, `AutoDistil` distills several compressed models with variable computational cost with the following major components.

## 3.1 Search Space Design

**Searchable transformer components.** From Transformers overview (Section 2) and our framework (Figure 2), we observe four important hyper-parameters for the Transformer blocks to include: (1) Feed-forward network (FFN) dimension - we encode this by the MLP (multi-layer perceptron) ratio defined as $r = \frac{d_f}{d_{hid}}$, with $d_f$ and $d_{hid}$ representing the intermediate dimension of the FFN and hidden dimension respectively; (2) Number of layers ($L$) to capture the network depth; (3) Hidden dimension ($d_{hid}$) to encode input; (4) Attention heads ($H$) for multi-head self-attention.

All of the above factors are important for model capacity and have a significant impact on the model size and computational cost. For instance, different layers have different feature representation capabilities. Recent works show that Transformer models are overparameterized [21, 22], such as the feed-forward layer (FFN), which is one of the most computation intensive components [23]. Therefore, we search for optimal MLP ratio and hidden dimension that reduce computational cost resulting from FFN layers. Furthermore, studies [24, 25] show that attention heads can be redundant when they learn to encode similar relationships for each word. Thus, we make the number of attention heads searchable as well.

**Inductive bias.** Prior works on CNNs [26] and Transformers [27] demonstrate that thinner and deeper neural networks with improved representation capacity perform better than wider and shallower ones. We incorporate this as an inductive bias to decide the number of layers to consider for the students in each of our $K$ sub-spaces (base, small, tiny), where we prefer deeper students in terms of the number of layers. Furthermore, we constrain all the Transformer layers in a given student model to share identical and homogeneous structures, i.e., the same number of attention heads, hidden dimension, etc. This not only reduces the size of the search space, it is also more friendly to hardware and software frameworks [13].

**Search space partition.** Existing works [13, 14] train a single large SuperLM containing millions of student architectures by weight-sharing. This leads to performance degradation due to optimization interference and convergence of subnetworks with very different sizes [11]. To mitigate such

interference, we employ a few-shot learning strategy [17, 16] as follows. We partition the whole Transformer search space into $K$ sub-spaces such that each sub-space covers different sizes of student models given by the number of parameters. For instance, $K = 3$ can cover typical student sizes, namely base, small and tiny versions. Table 2 shows the parameter ranges for the $K$ sub-spaces, along with the student configurations contained in each. Extensive details on the search space design can be found in the Appendix (Section D).

We now encode each sub-space into a SuperLM, where each student model in the space is a subnetwork of the SuperLM. Furthermore, all the student subnetworks share the weights of their common dimensions, with the SuperLM being the largest one in the search space. Considering $K$ independent SuperLMs, each one now has more capacity to encode a sub-space, in contrast to a limited capacity single SuperLM as in prior works. Furthermore, our choices for the heuristic partition and inductive bias result in less number of student models of comparable size in each sub-space which alleviates conflicts in weight-sharing.

We extract student subnetworks from the SuperLM by a simple truncation strategy like bottom-left extraction. In particular, given a specific architecture $\alpha = \{l, d_{hid}, r, h\}$, (i) we first extract alternate $l$ Transformer layers from the SuperLM; (ii) then extract bottom-left sub-matrices in terms of $d_{hid}$ and $r$ from the original matrices that represent the hidden dimension and the MLP ratio respectively; (iii) finally, for the attention heads, we extract the leftmost $h$ heads and retain the dimension

|  | SuperLM$_{\text{Tiny}}$ | SuperLM$_{\text{Small}}$ | SuperLM$_{\text{Base}}$ | BERT |
|---|---|---|---|---|
| #Subnets | 256 | 256 | 256 | N/A |
| #Layers | (4, 7, 1) | (9, 12, 1) | (9, 12, 1) | 12 |
| #Hid_dim | (128, 224, 32) | (256, 352, 32) | (544, 640, 32) | 768 |
| MLP Ratio | (2.0, 3.5, 0.5) | (2.5, 4.0, 0.5) | (2.5, 4.0, 0.5) | 4.0 |
| #Heads | (7, 10, 1) | (7, 10, 1) | (9, 12, 1) | 12 |
| #FLOPs | 40-367$M$ | 0.5-2.1$G$ | 2.1-7.9$G$ | 11.2$G$ |
| #Params | 4-10$M$ | 12-28$M$ | 39-79$M$ | 109$M$ |

Table 2: The search space of `AutoDistil` with $K$=3 partitions, each consisting of 256 subnets with variable computational cost. We train one SuperLM with weight-sharing for *each partition* with child models sharing transformer blocks. Each tuple represents the lowest value, highest value, and steps for each factor.

of each head as the SuperLM. This strategy is used during sampling sub-networks for SuperLM training via weight sharing after which the sub-network weights are updated via self-attention relation distillation. There can be better strategies to extract sub-networks (e.g., ordering heads by importance) that we defer to future work. We did explore some strategies for selecting layers (alternate vs. top vs. bottom) with results in Section B of Appendix.

## 3.2 Task-agnostic SuperLM Training

We illustrate SuperLM training process in Algorithm 1. Given a large pre-trained language model (e.g., BERT) as teacher, we initialize the SuperLM with the weights of teacher. In each step of SuperLM training, we randomly sample several student subnetworks from the search space; apply knowledge distillation between sampled subnetworks and the teacher to accumulate gradients; and then update the SuperLM. During sampling, we employ Sandwich rule [28], also used in BigNAS [11], that samples the smallest subnetwork, the largest subnetwork and $M$ random ones for updating SuperLM. The motivation is to improve the performance of all subnetworks by increasing the performance lower bound (smallest subnetwork) and upper bound (largest one) across all subnetworks.

We leverage deep self-attention distillation [4] for task-agnostic training. To this end, we employ multi-head self-attention relation distillation to align the attention distributions as well as scaled dot-product of keys, queries and values of the teacher and sampled student subnetworks. Consider $\mathbf{A}_1$, $\mathbf{A}_2$, $\mathbf{A}_3$ to denote the queries, keys and values of multiple relation heads of teacher model, and $\mathbf{B}_1$, $\mathbf{B}_2$, $\mathbf{B}_3$ respectively for a sampled subnetwork. Mean squared error (MSE($\cdot$)) between multi-head self-attention relation of teacher and sampled subnetwork is used as distillation objective:

$$\mathcal{L} = \sum_{i=1}^{3} \beta_i \mathcal{L}_i, \ \mathcal{L}_i = \frac{1}{H} \sum_{k=1}^{H} \text{MSE}(\mathbf{R}_{ik}^T, \mathbf{R}_{ik}^S), \tag{4}$$

where $\mathbf{R}_i^T = \text{softmax}(\mathbf{A}_i\mathbf{A}_i^\top/\sqrt{d_k})$, $\mathbf{R}_i^S = \text{softmax}(\mathbf{B}_i\mathbf{B}_i^\top/\sqrt{d_k})$, $H$ is the number of attention heads; $\mathbf{R}_i^T$ represents the teacher's $Q - Q$, $K - K$, or $V - V$ relation; $\mathbf{R}_i^S$ represents the same for student. $\mathbf{R}_{ik}^T$ is the relation information based on one attention head, and $d_k$ is the attention head size.

Relation knowledge distillation avoids the introduction of additional parameters to transform the student's representations with different dimensions to align to that of the teacher. For the teacher model and subnetworks with different number of attention heads, we first concatenate the self-attention vectors of different attention heads of the subnetwork and then split them according to the number of relation heads of the teacher model. Then, we align their queries with the same number of relation heads for distillation. In addition, we only transfer the self-attention knowledge from the last layer of the teacher model to the last layer of the student model.

The SuperLM for sub-space $\mathcal{A}_k$ is trained as:

$$\boldsymbol{W}_{\mathcal{A}_k}^* = argmin_{\boldsymbol{W}} \mathbb{E}_{\alpha \in \mathcal{A}}[\mathcal{L}(\boldsymbol{W}_\alpha; \boldsymbol{U}; \mathcal{D}_{train})], \tag{5}$$

where, $K$ is the number of sub-space partitions; $\boldsymbol{W}$ are the weights of the SuperLM; $\boldsymbol{W}_\alpha$ are the weights in $\boldsymbol{W}$ specified by the architecture $\alpha$; $\boldsymbol{U}$ are the weights of the teacher model including the self-attention module used for distillation; $\mathcal{D}_{train}$ is the training data set, and $\mathcal{L}(\cdot)$ is the self-attention loss function from Eqn. (4).

### 3.3 Lightweight Optimal Student Search

We outline two search strategies for selecting the optimal student subnetwork.

**Task-agnostic search.** We adopt this to be our primary strategy to compare against all baselines since it does not access any task label

---

**Algorithm 1** Few-shot Task-agnostic Knowledge Distillation with `AutoDistil`.

---

**Input:** Partitioned $K$ sub-spaces $\mathcal{A}_k$; initialized $K$ SuperLMs $S_k$ on $\mathcal{A}_k$; pre-trained teacher $T$; unlabeled data $D$; training epochs $E$; sampling steps $M$
**Output:** Trained SuperLMs $\{S_k\}$
**for** $k = 1$ **to** $K$ **do**
    **for** $i = 1$ **to** $E$ **do**
        Get a batch of data from $D$
        **for** $batch$ in $D$ **do**
            Clear gradients in SuperLM $S_k$
            **for** $m = 1$ **to** $M$ **do**
                Randomly sample a subnetwork $s$ from $S_k$
                Calculate self-attention distil. loss between subnetwork $s$ and teacher $T$ with Eqn. (4)
                Accumulate gradients
            **end for**
            Update $S_k$ with the accumulated gradients
        **end for**
    **end for**
**end for**

---

information. We compute the task-agnostic self-attention distillation loss for all student subnetworks using Eqn. (4) on a heldout validation set from the unlabeled training corpus. The student subnetworks are directly obtained by bottom-left extraction from the well-trained SuperLM (outlined in Section 3.1). This process is lightweight since it does not require any training or adaptation of the student and number of subnetworks is limited. The optimal student is given by the subnetwork with least validation loss subject to following constraint.

$$\alpha_{\mathcal{A}}^* = argmin_{\alpha \in \mathcal{A}_{1,2,\cdots K}} \mathcal{L}(\boldsymbol{W}_\alpha^*; \mathcal{D}_{val}), \quad s.t. \quad g(\alpha) < c, \tag{6}$$

where $\boldsymbol{W}_\alpha^*$ is the weights of architecture $\alpha$ obtained from $\boldsymbol{W}_{\mathcal{A}_k}^*$, $\mathcal{D}_{val}$ is the validation data set, $\mathcal{L}$ is the self-attention distillation loss, and $g(\cdot)$ is a function to calculate the computational cost (e.g., #FLOPs, #parameters) of the subnetwork subject to a given user-specified resource constraint $c$.

**Task-proxy search.** We compare our task-agnostic search against another strategy that considers a proxy task (e.g., MNLI [29]) with label information to fine-tune the 256 candidate subnetworks in each sub-space. The optimal student in each sub-space is given by the one with the best downstream task performance (e.g., accuracy). Note that, for this strategy, the proxy task is used only during search while the NAS training is still fully task-agnostic.

## 4 Experiments

**Datasets.** We conduct experiments on General Language Understanding Evaluation (GLUE) benchmark [30]. We compare our method with the baseline methods on two single-sentence classification tasks (CoLA [31], SST-2 [32]), two similarity and paraphrase tasks (MRPC [33], QQP [34]), and three inference tasks (MNLI [29], QNLI [35], RTE [36, 37, 38, 39])[3]. We report accuracy for MNLI, QNLI, QQP, SST-2, RTE, report f1 for MRPC, and report Matthew's correlation for CoLA.
**Baselines.** We compare against several *task-agnostic methods*[4] generating compressed models from BERT$_{base}$ teacher, using (i) knowledge distillation like BERT$_{SMALL}$ [40], Truncated BERT [29], DistilBERT [5], TinyBERT [6], MINILM [4]; as well as those based on Neural Architecture Search, like AutoTinyBERT [13], DynaBERT [12], and NAS-BERT [14].

---

[3]We ignore STS-B for a fair comparison with our strongest KD baseline MINILM [4] that do not report it.
[4]For a fair comparison, we do not include MobileBERT [7] that uses BERT$_{large}$ as teacher.

**AutoDistil configuration.** We use uncased $BERT_{BASE}$ as the teacher consisting of 12 Transformer layers, 12 attention heads; with the hidden dimension and MLP ratio being 768 and 4, respectively. It consists of $109M$ parameters with $11.2G$ FLOPs. We use English Wikipedia and BookCorpus data for SuperLM training with WordPiece tokenization. We use a batch size of 128 and $4e$-5 as the peak learning rate for 10 epochs. The maximum sequence length is set to 128. The coefficients in distillation objective (Eqn. (4)), $\beta_1$, $\beta_2$, and $\beta_3$, are all set to 1. We distill the self-attention knowledge of the last layer to train the SuperLM. Both the teacher and SuperLM are initialized with pre-trained $BERT_{BASE}$. Other hyper-parameter settings are shown in Appendix. We use 16 $V100$ GPUs to train the SuperLM with 336 GPU-hours as the training cost.

## 4.1 Finding the Optimal Compressed Models

**AutoDistil**$_{Agnostic}$ is obtained by fully task-agnostic training and task-agnostic search without using any task label information. We set a constraint in Eqn. (6) such that the #FLOPs of the optimal compressed model is atleast $50\%$ less than the teacher model. We rank all the subnetworks contained in all the partitions of the trained SuperLM by their self-attention distillation loss on the heldout validation set, and select the one that meets the constraint with the minimum loss.

**AutoDistil**$_{Proxy}$ uses MNLI [29] as a proxy to estimate downstream task performance of different subnetworks. Prior work [41] has demonstrated performance improvements in MNLI to be correlated to other GLUE tasks. To this end, we fine-tune all the 256 subnetworks in each partition of the trained superLMs, and select corresponding subnetworks with the best trade-off (more discussions on the optimal trade-off can be found in the Appendix (Section G)) between task performance (accuracy) and computational cost (#FLOPs). This results in $K$=3 optimal students, corresponding to AutoDistil$_{Proxy_B}$, AutoDistil$_{Proxy_S}$ and AutoDistil$_{Proxy_T}$ obtained from the corresponding sub-spaces of SuperLM$_{Base}$, SuperLM$_{Small}$ and SuperLM$_{Tiny}$, respectively. Notably all students are obtained from the AutoDistil SuperLM still trained in a fully task-agnostic fashion.

### 4.1.1 Comparison with Traditional Knowledge Distillation Baselines

We compare AutoDistil against state-of-the-art KD models distilled from the same teacher $BERT_{BASE}$ in Table 3 with respect to the following measures: computational cost in the form of (i) FLOPs and (ii) parameters, along with (iii) improvement in the average task performance aggregated over all the GLUE tasks. We observe that the compressed model AutoDistil$_{Agnostic}$ generated via our task-agnostic SuperLM training leads to upto $3x$ reduction in FLOPs over state-of-the-art distilled models (e.g., MINILM [4], TinyBERT [6], DistilBERT [5]) that are hand-engineered while matching the overall task performance. The most aggressive compressed version corresponding to AutoDistil$_{Proxy_T}$ obtains a massive $41x$ reduction in FLOPs over $BERT_{BASE}$ while incurring 5 point accuracy drop in GLUE (excluding CoLA) and 10 point drop (including CoLA). Notably CoLA is a syntactic task in contrast to other semantic tasks in the benchmark like natural language inference, paraphrase detection and sentiment classification. This depicts an interesting impact of massive model compression on varying task types.

### 4.1.2 Comparison with Neural Architecture Search Baselines

We report the performance of several NAS-generated student models of comparable FLOPs and parameters from corresponding papers in Table 3. AutoDistil outperforms all competing methods on aggregate for all sizes; except for small-sized model; where it has marginally lower performance (0.1 points on avg) compared to AutoTinyBERT. It is worthwhile to note that computational cost of training process is another important dimension for comparing methods. This is especially important when comparing to NAS methods that use multi-stage training; where additional pre-training and distillation is applied to NAS-generated candidates.

To better understand the impact of single-stage vs. multi-stage methods on the training cost, we

| Cost (GPU hours) | AutoTiny BERT | AutoTiny BERT-Fast | Auto Distil |
|---|---|---|---|
| SuperNet Training | *NR* | *NR* | 336 |
| Search | 150 | 12 | <1 |
| Further Training | 870 | 290 | 0 |

Table 4: Training cost (V100 GPU hours) comparison for generating students of similar FLOPs. $^{NR}$AutoTiny- BERT does not report the cost of SuperNet training - typically the most expensive step. Further Training refers to additional pre-training applied to NAS-generated candidates.

Table 3: Performance comparison between students from traditional task-agnostic distillation; multi-stage one-shot NAS with additional pre-training, distillation; and single-stage few-shot `AutoDistil` Our results are averaged over 5 runs with baselines reported from corresponding papers.

| Model (Metric) | #FLOPs (G) | #Para (M) | MNLI-m (Acc) | QNLI (Acc) | QQP (Acc) | SST-2 (Acc) | CoLA (Mcc) | MRPC (Acc) | RTE (Acc) | Average |
|---|---|---|---|---|---|---|---|---|---|---|
| BERT$_{\text{BASE}}$ [1] (teacher) | 11.2 | 109 | 84.5 | 91.7 | 91.3 | 93.2 | 58.9 | 87.3 | 68.6 | 82.2 |
| *Base-sized Models from Task-agnostic KD Methods and* `AutoDistil` | | | | | | | | | | |
| BERT$_{\text{SMALL}}$ [40] | 5.66 | 66.5 | 81.8 | 89.8 | 90.6 | 91.2 | 53.5 | 84.9 | 67.9 | 80.0 |
| Truncated BERT [29] | 5.66 | 66.5 | 81.2 | 87.9 | 90.4 | 90.8 | 41.4 | 82.7 | 65.5 | 77.1 |
| DistilBERT[5] | 5.66 | 66.5 | 82.2 | 89.2 | 88.5 | 91.3 | 51.3 | 87.5 | 59.9 | 78.6 |
| TinyBERT [6] | 5.66 | 66.5 | 83.5 | 90.5 | 90.6 | 91.6 | 42.8 | 88.4 | 72.2 | 79.9 |
| MINILM [4] | 5.66 | 66.5 | 84.0 | 91.0 | 91.0 | 92.0 | 49.2 | 88.4 | 71.5 | 81.0 |
| `AutoDistil`$_{\text{ProxyB}}$ | 4.40 | 50.1 | 83.8 | 90.8 | 91.1 | 91.1 | 55.0 | 88.8 | 71.9 | 81.7 |
| *Small-sized Models from Multi-stage One-shot NAS Methods and* `AutoDistil` | | | | | | | | | | |
| AutoTinyBERT-KD-S1 [13] | 1.69 | 30.0 | 82.3 | 89.7 | 89.9 | 91.4 | 47.3 | 88.5 | 71.1 | 80.0 |
| DynaBERT [12] | 1.81 | 37.7 | 82.3 | 88.5 | 90.4 | 92.0 | 43.7 | 81.4 | 63.2 | 77.4 |
| NAS-BERT$_{10}$ [14] | 2.30 | 10.0 | 76.4 | 86.3 | 88.5 | 88.6 | 34.0 | 79.1 | 66.6 | 74.2 |
| `AutoDistil`$_{\text{ProxyS}}$ | 2.02 | 26.1 | 83.2 | 90.0 | 90.6 | 90.1 | 48.3 | 88.3 | 69.4 | 79.9 |
| `AutoDistil`$_{\text{Agnostic}}$ | 2.13 | 26.8 | 82.8 | 89.9 | 90.8 | 90.6 | 47.1 | 87.3 | 69.0 | 79.6 |
| *Tiny-sized Models from Multi-stage One-shot NAS Methods and* `AutoDistil` | | | | | | | | | | |
| AutoTinyBERT-KD-S4 [13] | 0.30 | 10.1 | 76.0 | 85.5 | 86.9 | 86.8 | 20.4 | 81.4 | 64.9 | 71.7 |
| NAS-BERT$_5$ [14] | 0.87 | 5.00 | 74.4 | 84.9 | 85.8 | 87.3 | 19.8 | 79.6 | 66.6 | 71.2 |
| `AutoDistil`$_{\text{ProxyT}}$ | 0.27 | 6.88 | 79.0 | 86.4 | 89.1 | 85.9 | 24.8 | 78.5 | 64.3 | 72.6 |

compare the overall cost of NAS for `AutoDistil` and that reported in AutoTinyBERT[5] for the small model segment in Table 4. AutoDistil is much cheaper due to its single-stage training protocol; where no additional pre-training or distillation is needed. It is worth noting that the overall SuperNet training cost of `AutoDistil` (the most expensive component of NAS) is less or comparable to the additional training cost of re-training candidate models for AutoTinyBERT. Note that AutoTinyBERT does not report their SuperNet training cost. Additionally, `AutoDistil` has a much faster search mechanism due to (1) inductive biases built into the search space definition to limit the number of student architectures and (2) task-agnostic search that only requires computing self-attention validation loss without the need for any training.

Finally, we show the pareto frontier of student subnetworks generated by several KD and NAS methods in Figure 3 for the MNLI task. The blue points represent all the subnetworks extracted from `AutoDistil` and red points denote the optimal ones, all fine-tuned on the MNLI task. We observe the optimal `AutoDistil` models to outperform several competing methods.

### 4.1.3 Task-agnostic Training Strategies

We study different task-agnostic strategies for SuperLM training in `AutoDistil`. Specifically, we compare three strategies in Table 5. (i) We replacing the KD loss in Eqn. (4) with masked language modeling (MLM) loss [1] to calculate gradients which is the most widely used task-agnostic pre-training and distillation strategy. (ii) KD$_{att}$+Cont further continues training the searched compressed models on the large language corpus. (iii) KD$_{att}$ is the strategy adopted in `AutoDistil`

Table 5: Comparing task-agnostic SuperLM training strategies.

| Strategy | MRPC | RTE | MNLI |
|---|---|---|---|
| MLM | 89.4 | 68.2 | 82.2 |
| KD$_{att}$+Cont. | 91.0 | 71.8 | 83.5 |
| KD$_{att}$ | 91.2 | 71.5 | 83.2 |

for self-attention distillation. We evaluate subnetworks with the same architecture (6 layers, 768 hidden, 12 heads, MLP ratio 4) from the trained SuperLM. We fine-tune subnetworks on MRPC, RTE, and MNLI tasks, and report f1, accuracy, and accuracy, respectively. MRPC and RTE are paraphrase detection and natural language inference tasks, respectively. They are low-resource tasks with limited training examples ( $3K$ labels each) for fine-tuning. This allows us to evaluate transferability of the compressed models trained on unlabeled general domain (e.g., Wikipedia) and fine-tuned on downstream tasks with limited labels. MNLI has a larger number of examples

---

[5]Other NAS methods either use a different hardware for training or do not report the cost.

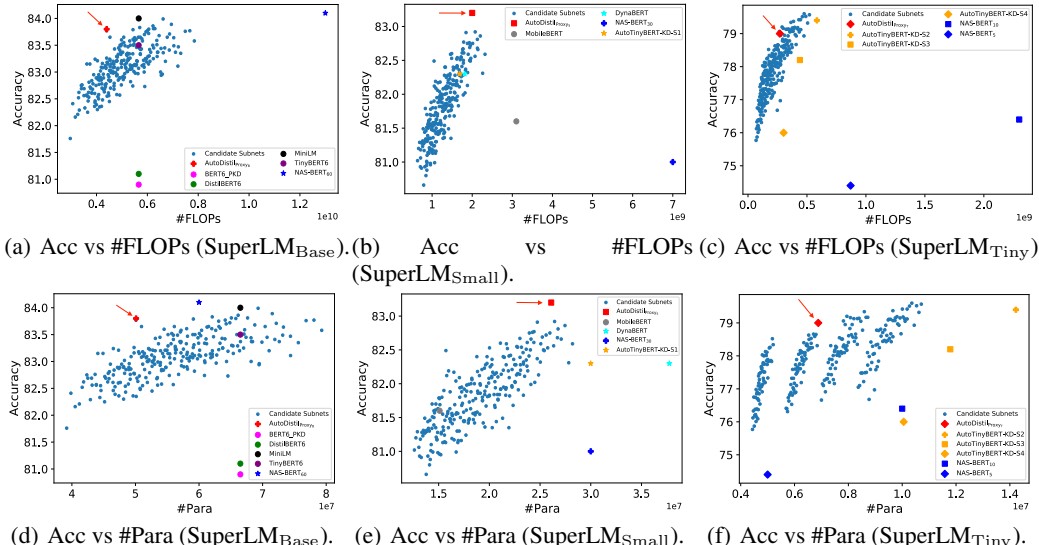

(a) Acc vs #FLOPs (SuperLM$_{Base}$). (b) Acc vs #FLOPs (SuperLM$_{Small}$). (c) Acc vs #FLOPs (SuperLM$_{Tiny}$).

(d) Acc vs #Para (SuperLM$_{Base}$). (e) Acc vs #Para (SuperLM$_{Small}$). (f) Acc vs #Para (SuperLM$_{Tiny}$).

Figure 3: Computational cost vs. task (MNLI) performance trade-off for all 256 subnetworks contained in each of $K$ SuperLMs (base, small and tiny). 3(a)-3(c) show the trade-off between accuracy (Y-axis) and #FLOPs (X-axis), and 3(d)-3(f) show the trade-off between accuracy (Y-axis) and #Para (X-axis) with optimal compressed `AutoDistil` student for each SuperLM in red.

($400K$). Prior work [41] has shown MNLI to transfer well to other tasks in GLUE. First, we observe self-attention distillation to perform better than MLM, for SuperLM training. Second, we observe limited performance gains with continued training of the optimal subnetworks from NAS as done in existing works **demonstrating the effectiveness of our single-stage training protocol**.

#### 4.1.4 One-shot vs. Few-shot NAS with Varying $K$

For few-shot NAS, we choose $K$=3 (i.e. 3 subspaces) for following reasons: (1) 3 sub-spaces correspond to base, small and tiny model sizes; as used in prior work in CV; e.g. AutoFormer [17], (2) searching over different values of $K$ is resource-extensive since it requires training $K$ SuperLMs for each choice of $K$, (3) As $K$ increases, search process becomes similar to the undesirable brute-force discrete search that trains all models in search space individually.

Table 6: Search space design strategies.

| Task | Search Space Size (#subnetworks) | | | |
|------|------|------|------|------|
| | One-shot ($K = 1$) | | | Few-shot ($K = 3$) |
| | 27 | 864 | 11232 | 256*3 |
| MRPC | 88.2 | 87.5 | 85.1 | 91.2 |
| RTE | 67.2 | 64.5 | 62.8 | 71.8 |

To understand the effect of few-shot vs. one-shot NAS, we compare the performance of a single space ($K = 1$) to multiple sub-spaces ($K = 3$). We extract subnetworks with same architecture (6 layers, 768 hidden, 12 heads, MLP ratio 4) from trained SuperLMs for each strategy for evaluation with results in Table 6. For one-shot NAS, we consider a single search space containing varying number of subnetworks (e.g., 27, 864, 11232). Few-shot NAS contains 256 subnetworks in each partition. We fine-tune subnetworks on RTE and MRPC tasks, and report accuracy and f1 respectively. We observe fewer subnetworks in a single search space for one-shot NAS result in better performance. This results from optimization interference and gradient conflicts as the number and size of subnetworks increase in the space. Finally, our design strategy performs the best while containing lesser number of subnetworks **demonstrating the benefit of few-shot NAS for language model distillation**.

#### 4.1.5 Comparing Search Strategies and Optimal Architectures

From Table 3, we observe that the student models `AutoDistil`$_{Agnostic}$ and `AutoDistil`$_{Proxy_S}$ obtained from SuperLM$_{small}$ by task-agnostic and task-proxy search strategies respectively obtain a similar trade-off between performance and cost. The task-proxy search results in a minor performance gain 0.3 over the fully task-agnostic search mechanism. Table 7 shows the configuration of searched optimal architectures from `AutoDistil` with corresponding computational cost. For reference, we also show the architecture of the teacher BERT$_{BASE}$ and a state-of-the-art distilled model MINILM [4] that are hand-engineered. We observe that the obtained architectural hyper-parameters are quite

non-standard and difficult to obtain by trial and error considering the large space of Transformer architectures. We also observe that optimal compressed models have thin-and-deep structure consistent with findings that thinner and deeper models perform better [26] than wider and shallower ones. While we use this as an inductive bias for sub-space partitioning, our search space (Table 2) also contains diverse subnetworks with different depth and width. Non-maximal MLP ratio and attention heads for optimal compression indicate that self-attention and feed-forward layers of Transformers are overparameterized [21, 22].

Table 7: Architecture comparison between the optimal compressed students searched by `AutoDistil` with state-of-the-art hand-engineered students distilled from $BERT_{BASE}$.

| Model | #Layers | #Hid | Ratio | #Heads | #FLOPs | #Para |
|---|---|---|---|---|---|---|
| $BERT_{BASE}$ | 12 | 768 | 4 | 12 | 11.2G | 109M |
| MINILM | 6 | 768 | 4 | 6 | 5.66G | 66.5M |
| `AutoDis.`$_{Agnostic}$ | 11 | 352 | 4 | 10 | 2.13G | 26.8M |
| `AutoDis.`$_{Proxy_B}$ | 12 | 544 | 3 | 9 | 4.40G | 50.1M |
| `AutoDis.`$_{Proxy_S}$ | 11 | 352 | 4 | 8 | 2.02G | 26.1M |
| `AutoDis.`$_{Proxy_T}$ | 7 | 160 | 3.5 | 10 | 0.27G | 6.88M |

## 5  Related Work

**Task-specific knowledge distillation.** Knowledge distillation (KD) [42] is a widely used technique for model compression, which transfers knowledge from a large teacher to a smaller student. Task-specific KD aims to generate smaller students by using downstream task label information. Typical task-specific KD works include BERT-PKD [43], $BERT_{SMALL}$ [40], TinyBERT [6], DynaBERT [12], and SparseBERT [44]. While task-specific KD often achieves good task performance, a typical drawback is that it is resource-consuming to run KD for each and every task, and also not scalable.
**Task-agnostic knowledge distillation.** In contrast to task-specific KD, we explore task-agnostic KD that does not use any task label information. The distilled task-agnostic models can be re-used by simply fine-tuning on downstream tasks. Task-agnostic KD leverages knowledge from soft target probabilities, hidden states, layer mappings and self-attention distributions of teacher to train student models. Typical task-agnostic KD works include DistilBERT [5], MobileBERT [7], and MiniLM [4]. MobileBERT assumes that students have the same number of layers as the teacher for layer-by-layer distillation. MiniLM transfers self-attention knowledge from the last layer of the teacher to that of the student. These works rely on hand-designed architecture for the student models for KD that requires several trials, and needs to be repeated for a new student with a different cost. In contrast, we develop techniques to automatically design and distill several student models with variable cost using NAS.
**Neural Architecture Search.** While NAS has been extensively studied in computer vision [8, 9, 10, 11], there has been relatively less exploration in natural language processing. Evolved Transformer [45] and HAT [46] search for efficient sub-networks from the Transformer architecture for machine translation tasks. Some recent approaches closest to our method include, DynaBERT [12], AutoTinyBERT [13] and NAS-BERT [14]. DynaBERT performs task-specific distillation. NAS-BERT performs two-stage knowledge distillation with pre-training and fine-tuning of candidates. Similar to above approaches, AutoTinyBERT also employs one-shot NAS with a single large search space containing millions of subnetworks that result in co-adaption and weight-sharing challenges for SuperLM training. Further it also uses a multi-stage training protocol for further pre-training and distillation of the NAS-generated candidates. In contrast, `AutoDistil` employs few-shot NAS with a compact search space design with a single-stage task-agnostic training protocol. This further allows us to do a lightweight search for the optimal student without re-training. [16] studies few-shot task-specific NAS in the CV domain with CNN architectures, and different design and search spaces from `AutoDistil`. More comparisons between them can be found in the Appendix (Section A.2).

## 6  Conclusion

We develop a few-shot task-agnostic NAS method, namely `AutoDistil` to distil large language models into compressed students with variable computational cost. To address the co-adaption and weight-sharing challenges for SuperLM training, we partition the Transformer search space into $K$ compact sub-spaces covering important architectural components like its depth, width, and attention heads. We leverage self-attention distillation for fully task-agnostic SuperLM training and lightweight optimal search without any re-training. Obtained students can be re-used by simply fine-tuning on tasks. `AutoDistil` generates students with $3x$ less computational cost (FLOPs) than state-of-the-art task-agnostic KD methods while obtaining a similar task performance in GLUE.

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
