# Appendix

## A  Comparisons with Existing NAS and KD Methods

### A.1  Training Cost Comparisons with Traditional Task-agnostic KD Methods

Different hardwares (e.g., FPGA, CPU, GPU) have different resource constraints. AutoDistil generates a gallery of fully trained compressed student models with variable resource constraints (e.g., FLOPs, parameters) using NAS. One can simply choose a model from the trained pool given the resource constraint and only fine-tune on the downstream task. In contrast, traditional task-agnostic knowledge distillation (KD) methods (e.g., MiniLM) target specific compression rate and needs to be trained repeatedly for different student configurations (corresponding to different resource constraints). Therefore, AutoDistil has a much reduced amortized computation cost even considering traditional KD methods. Further, traditional methods require several trial and errors to come up with a viable candidate architecture given a constraint before running the KD algorithm.

In practice, additional or continued training of the optimal student architecture has demonstrated increased task performance with increased computational cost as in AutoTinyBERT (cost comparison in Table 4). The major advantage of AutoDistil is a single stage training scheme without additional training. We perform an ablation in Table 5 where we continue training the searched model with self-attention distillation for additional steps referred as 'KD$_{att}$+Cont.' (similar to MiniLM). But we did not observe any significant gains on a subset of the tasks.

### A.2  Comparisons between AutoDistil and Few-shot CV NAS [16]

**Fully task-agnostic SuperNet training**. AutoDistil training is fully task-agnostic in contrast to [16] that uses task-specific NAS. Task-agnostic NAS is challenging since we do not have access to task labels during training and we want to show generalization on evaluating diverse downstream NLU tasks in the GLUE benchmark. AutoDistil leverages self-attention distillation that is an unsupervised training objective. Incorporating self-attention loss for training and distillation in NAS is non-trivial as it requires aligning attention states of diverse student subnetworks and the large teacher model. We develop an extraction and alignment strategy (Section 3.2) to address this challenge.

**NLP vs. CV domain**. AutoDistil works on the NLP domain with the Transformer architecture (see Figure 2 under the pre-training and fine-tuning paradigm, while reference [16] works on the CV domain with a CNN architecture with different design and search spaces. Different from CV domain, NLP tasks have different objectives and evaluation metrics for classification (e.g., MNLI), regression (e.g., STS-B) and correlation (e.g., CoLA). Overall, the search space design (Section 3.1), SuperNet training with distillation and sub-network extraction strategy (Section 3.2) and search strategy (Section 3.3) are all quite different.

### A.3  Comparisons between AutoDistil and DynaBERT [12]

Compared with DynaBERT, (i) `AutoDistil` search space is more fine-grained. For instance, we independently search for width, depth, heads, MLP ratio etc. as opposed to searching for a constant depth ($m_d$) or width multiplier ($m_w$) in DynaBERT which only considers 12 possible combinations of $m_d$ and $m_w$; (ii) our training objective does not require labels and is fully task-agnostic with subnetwork attention state alignment for self-attention relation distillation; (iii) further, AutoDistil uses few-shot NAS (Table 1) to mitigate gradient conflicts in SuperNet training, while DynaBERT applies one-shot NAS; (iv) DynaBERT uses additional tricks like data augmentation and teacher assistant also specific to each task, whereas AutoDistil uses a single-stage task-agnostic training resulting in reduced computational cost.

### A.4  Additional Quantitative Comparisons with Existing NAS and KD Methods

We compare different `AutoDistil` compressed models against state-of-the-art KD and NAS models distilled from the same teacher BERT$_{BASE}$. We present the relative performance improvement of `AutoDistil` over several baselines in Table 8 with respect to the following measures: savings in computational cost in the form of (i) FLOPs and (ii) parameter reduction, along with (iii) improvement in the average task performance aggregated over all the GLUE tasks.

Table 8: Performance comparison between models distilled by `AutoDistil` against several task-agnostic students (6 layer, 768 hidden size, 12 heads) distilled from $BERT_{BASE}$. We report the relative reduction in computational cost (#FLOPs and #Parameters) and improvement in average task performance on GLUE (dev) over all baselines. $AutoDistil_{Agnostic}$ is obtained by task-agnostic search. $AutoDistil_{Proxy_B}$ and $AutoDistil_{Proxy_S}$ are obtained by task-proxy search from $SuperLM_{base}$ and $SuperLM_{small}$ respectively.

| Model | $AutoDistil_{Agnostic}$ | | | $AutoDistil_{Proxy_B}$ | | | $AutoDistil_{Proxy_S}$ | | |
|---|---|---|---|---|---|---|---|---|---|
| | $\Delta$FLOPs | $\Delta$Para | $\Delta$Avg. | $\Delta$FLOPs | $\Delta$Para | $\Delta$Avg. | $\Delta$FLOPs | $\Delta$Para | $\Delta$Avg. |
| $BERT_{BASE}$ [1] (teacher) | 81.1% | 75.5% | -2.6 | 60.9% | 54.3% | -0.5 | 82.0% | 76.2% | -2.3 |
| $BERT_{SMALL}$ [40] | 62.4% | 59.7% | -0.3 | 22.3% | 24.7% | +1.8 | 64.3% | 60.8% | -0.02 |
| Truncated BERT [29] | 62.4% | 59.7% | +2.5 | 22.3% | 24.7% | +4.6 | 64.3% | 60.8% | +2.8 |
| DistilBERT[5] | 62.4% | 59.7% | +1.1 | 22.3% | 24.7% | +3.2 | 64.3% | 60.8% | +1.4 |
| TinyBERT [6] | 62.4% | 59.7% | -0.3 | 22.3% | 24.7% | +1.8 | 64.3% | 60.8% | +0.0 |
| MINILM [29] | 62.4% | 59.7% | -1.4 | 22.3% | 24.7% | +0.7 | 64.3% | 60.8% | -1.1 |

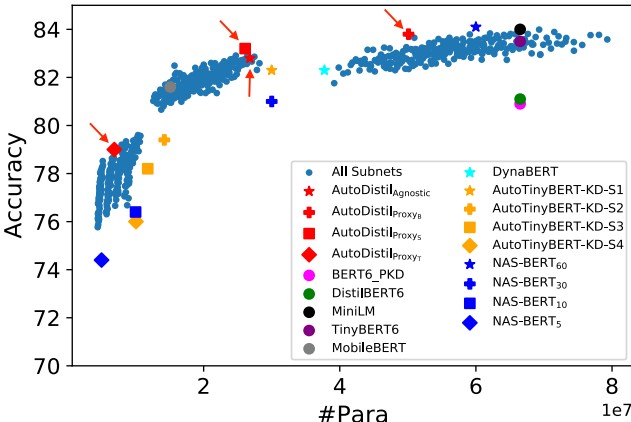

Figure 4: Comparison between `AutoDistil` and state-of-the-art distilled models.

We also compare `AutoDistil` with state-of-the-art distilled models in terms of the trade-off between model size (#Para) and performance (accuracy). The results are shown in Figure 4. `AutoDistil` uses few-shot task-agnostic Neural Architecture Search to distill several compressed students with variable #Para (x-axis) from $K$=3 SuperLMs (corresponding to each point cloud) trained on $K$ sub-spaces of Transformer search space. Each student extracted from the SuperLM is fine-tuned on MNLI with y-axis showing accuracy. The best student from each SuperLM is marked in red. Given any state-of-the-art distilled model, `AutoDistil` generates a better candidate with less #Para and improved task performance from corresponding search space.

## B  Layer Selection Strategies

Table 9: Effects of layer selection strategies.

| Strategy | MRPC | RTE |
|---|---|---|
| Alternate_Dropping | 91.2 | 71.8 |
| Top_Dropping | 90.6 | 68.5 |
| Alternate_Top_Dropping | 85.7 | 62.7 |

We study different strategies to construct subnetwork layers by selecting layers from the superLM model. Alternate_Dropping is the strategy adopted in `AutoDistil` such that we drop alternating odd layers from the superLM model to construct subnetwork layers. Top_Dropping means that we drop top layers of superLM to construct subnetwork layers. Alternate_Top_Dropping means that we first perform Alternate_Dropping in superLM training stage and then perform Top_Dropping in fine-tuning stage (please refer to [47] for more details of different layer selection strategies). For all strategies, we perform knowledge distillation between the last layer of the teacher model and the last layer of the subnetworks. We evaluate the subnetworks with the same architecture (#layer=6, #hid=768, R=4, #heads=12) after superLM is trained. We report accuracy and f1 for RTE and MRPC, respectively.

We report the results in Table 9. We observe that the strategy of Alternate_Dropping achieves the best performance on both MRPC and RTE tasks, which demonstrates the effectiveness of the layer selection strategy used in `AutoDistil`. Alternate_Top_Dropping performs the worst due to interference when different layer selection strategies are used in the superLM training stage and the fine-tuning stage of compressed models. This indicates that the knowledge contained in the superLM model and the compressed model is structured and that it is non-trivial to select layers from superLM to extract subnetwork layers.

## C Scaling of Training Data

Table 10: Scaling of training data.

| Strategy | MNLI (393k) | ParaNMT (5M) | Wiki (29M) | Wiki+Book (40M) |
|---|---|---|---|---|
| MRPC | 88.3 | 88.2 | 89.4 | 91.2 |
| RTE | 65.4 | 67.2 | 68.6 | 71.8 |

We investigate the effects of data sets of different sizes used for superLM training. In particular, we compare MNLI [29], ParaNMT [48] (we sampled 5 million samples from the original 50 million data), Wiki, and Wiki+Book [49]. We report the size of each data set and the performance of `AutoDistil` with each training data set in Table 10. We observe that `AutoDistil` performs the best with Wiki+Book data set, and the larger the data set, the better the performance. Moreover, we observe similar performance for MNLI and ParaNMT data sets, especially on MRPC task. This is because MNLI is correlated to other GLUE tasks. In addition, we observe that an increase in the amount of data does not guarantee to bring an equivalent increase in performance. For example, Wiki data set is more than five times larger than ParaNMT data set, but our method performs only about 1% better With Wiki data set than with ParaNMT. These observations illustrate that while using a larger data set does improve the performance of the method, the improvement could be quite limited.

## D Search Space Design

In general, we partition the whole Transformer search space into $K = 3$ sub-spaces such that each sub-space covers different sizes of student models (by number of parameters) depicting Tiny, Small and Base model sizes. Given a BERT-sized teacher model ($109M$ params), we roughly set the partition thresholds for Tiny, Small and Base sizes at $10M$, $40M$ and $80M$ params. From Table 2 (#Params row), we observe that each partition contains compressed models from prior work – allowing us to fairly compare the models in each partition on accuracy vs. params/FLOPs.

For our search space, each partition still contains thousands of candidate subnetworks not all of which are useful. Now, we leverage two primary heuristics: (i) we constrain all layers in a sampled student subnetwork to be homogeneous i.e., the same number of attention heads, hidden dimension, etc. This not only reduces the search space, it is also more friendly to hardware and software frameworks [13]. (ii) Motivated by previous work [26, 27] showing that thinner and deeper neural networks have better representation capabilities and perform better than wider and shallower neural networks, we designed sub-spaces with deeper layers (e.g., $4 - 7$ for Tiny, $9 - 12$ for Small and Base) and compute the range of hidden dimensions to meet the overall model parameter budget in each partition. Additional

constraints arise from Transformer design principles, for instance, hidden size is always a multiple of the number of attention heads. While the above steps require enumeration of different subnetwork architectures, this is typically fast given an algebraic expression to compute model parameters as a function of layers, heads, hidden size, etc. (included in source code), does not require any training, and a one-time process depending only on the teacher model architecture.

The impact of network depth on model performance has been observed with both convolution architectures [26] and Transformers [27]. Figure 2(b) in [27] shows the impact of Transformer depth on MNLI accuracy given similar overall model parameters.

## E    Transferability of Optimal Architectures

The transferability of optimal architectures has been studied with regards to model pruning in the lottery ticket hypothesis work [41] for BERT. They observe that transferability seems to correlate with the number of training examples available for the source task. This is particularly beneficial with MNLI containing a large number of training examples as compared to other low-resource tasks in the GLUE benchmark. Similar to [41], we also observe MNLI to transfer well to other tasks in the GLUE benchmark with AutoDistil-proxy even outperforming task-specific NAS method like DynaBERT (Table 3) on both parameters ($26.1M$ vs. $37.7M$) and average accuracy (79.9 vs. 77.4).

In general, a better teacher model leads to a better student model [50, 4] during distillation. We adopted BERT as teacher for a fair comparison with existing works with the same teacher. We also compare with different training objectives like self-attention distillation and masked language modeling (Table 5) and demonstrate the former to work better for our SuperNet training. We demonstrate transferability by training the AutoDistil students in a task-agnostic manner and evaluating on different downstream tasks (Table 3). Note that these tasks are quite diverse ranging from classification (e.g., MNLI), regression (e.g., STS-B) and correlation (e.g., CoLA). We also demonstrate this to work better or comparable to task-specific NAS methods (e.g., DynaBERT, AutoTinyBERT) with further reduction of computational cost.

## F    Application of Few-shot Task-agnostic NAS Method to Other Domains

Most NAS works in computer vision (CV) (e.g., Once-for-all, One-Shot NAS) leverage hard class labels from a given task (e.g., image classification). They often use similar training recipes for SuperNets as in ImageNet-trained models (e.g., MobileNet, RegNet) for task-specific optimization with accuracy as an evaluation metric. In contrast, the few-shot task-agnostic NAS strategy used in AutoDistil training is fully task-agnostic and does not access task labels during SuperNet training.

A potential method to adopt this strategy for CV domain is to consider a self-supervised learning framework like SimCLR [51] that leverages data augmentation for consistency learning. This requires both a self-supervised teacher like SimCLR and a self-supervised training objective (e.g., self-attention relation distillation for Transformers or architecture-agnostic consistency learning). This forms an interesting direction for future work.

## G    Quantifying "Best Trade-off" between Task Performance (e.g., Accuracy) and Computational Cost (e.g., #FLOPs)

We describe how to search for the optimal sub-network in Section 3.3 and Section 4.1. The "best trade-off" for optimal student selection is given by the sub-network with the least validation loss subject to the resource constraint as described in Eqn (6). For instance, we set a constraint in Eqn. (6) such that the #FLOPs of the optimal Base-sized task-agnostic compressed model is atleast 50% less than the teacher model. Since the SuperNet training is task-agnostic, the obtained student models have to be fine-tuned on downstream tasks to report the final task performance (similar to pre-train and fine-tune paradigm of BERT-like language models).

## H   Why Are the Selected Models (Red in Figure 1) Not Always the Best Performing Models?

Note that our objective is to minimize the #FLOPs and maximize the accuracy (e.g., on MNLI) with the trade-off determined by the resource constraint for different partitions (Base, Small, Tiny). Given a gallery of compressed models from AutoDistil with variable FLOPs and performance, we use strategy from Section G for optimal model selection.

Another potential reason why red is always not the best model is that we use the heldout validation set from the unlabeled training corpus (Wikipedia + BooksCorpus) for student selection and then evaluate them on MNLI (see Sections 3.3 and 4.1) which may not be optimal due to sample differences in the two datasets.

## I   Subnetwork Evaluation Strategy in Task-agnostic Search

The validation set contains $300K$ instances. We use $128$ as the sequence length and batch size. (ii) Table 4 reports the search cost for Small-sized models from AutoDistil and AutoTinyBERT. Note that this step does not require any training for AutoDistil. We compute only the self-attention relation loss for all the 256 student subnetworks ($5x - 22x$ speedups for Small-sized models) using Equation 4 with the teacher relations computed only once. We use Equation 6 to select the subnetwork with desired trade-off with deterministic computation of the FLOPs. The algebraic expression to compute FLOPs as a function of layers, heads, hidden size etc. is included in the submitted source code. In contrast, AutoTinyBERT performs task-specific search which requires fine-tuning the subnetworks on the task (e.g., MNLI) thereby increasing the search cost.

## J   Hyper-parameter Settings for Fine-Tuning

Table 11: Hyper-parameters used for fine-tuning on GLUE.

| Tasks | Learning Rate | Batch Size | Epochs |
|-------|---------------|------------|--------|
| MNLI-m | 2e-5 | 32 | 5 |
| QNLI | 2e-5 | 32 | 5 |
| QQP | 2e-5 | 32 | 5 |
| SST-2 | 2e-5 | 32 | 10 |
| CoLA | 1e-5 | 32 | 20 |
| MRPC | 2e-5 | 32 | 10 |
| RTE | 2e-5 | 32 | 10 |

We report the fine-tuning hyper-parameter settings of GLUE benchmark in Table 11. `AutoDistil` and baselines follow the same settings.

## K   Limitations and Broader Impact

In this work, we introduce a framework for distilling large pre-trained neural language models with resource constraints.

This work is likely to increase the progress of NLP applications and drive the development of general-purpose language systems for deployment environments with limited resources. Our framework can be used for applications in finance, legal, healthcare, retail and other domains for edge scenarios where adoption of deep neural networks have been hindered due to resource efficiency concerns.

In our work, we propose a solution of a single-stage training combining NAS and distillation with no further pre-training or augmentation, and experiments on GLUE benchmark demonstrate our solution to outperform state-of-the-art KD and NAS methods with upto 3x additional reduction in computational cost and negligible loss in task performance.

While our framework accelerates the progress of NLP, it may also suffer from similar concerns as with the use of large pre-trained models by malicious agents for propagating bias, misinformation and indulging in other nefarious activities.