# OpenReview forum: "Few-shot Task-agnostic Neural Architecture Search for Distilling Large Language Models"
_NeurIPS.cc/2022/Conference — NeurIPS 2022 Accept_

### Official Review · Reviewer_ov4n · 2022-07-04

**Rating:** 4
**Confidence:** 2
**Soundness:** 3 good
**Presentation:** 2 fair
**Contribution:** 2 fair

**Summary:**

The paper preposes AutoDistill, a recipe for distilling large language models into smaller sizes when a budget is given.

AutoDistill has three major steps:
* designing the search space with 4 hyperparameters, and partition the large search space into three smaller ones (K=3; tiny, small, base).
* training a SuperLM in each partition, with a task-agnostic objective similar to MiniLM (Wang et al., 2020)
* obtain all possible student architectures by pruning the SuperLM, and find the best one by enumerating the model and computing the loss on either validation data or task-specific data

AutoDistill achieves superior performance on GLUE benchmark when compared to baselines such as (a) fine-tuning, (b) task-agnostic distillation baselines: DistilBERT, TinyBERT, and (c) NAS baselines: NAS-BERT. Meanwhile AutoDistill is more computationally efficient as demonstrated in Table 3.


**Questions:**

(Disclaimer: I know little about NAS or one-shot/few-shot NAS. My ratings are based on my current understanding and I hope the authors can clarify the following for me)

* Does "few-shot" in the method mainly mean splitting the search space into K=3 subspaces?
* What is the major difference between AutoDistill and Reference [16] (Zhao et al., 2021)?
* The motivation of few-shot NAS is to alleviate conflicts in weight-sharing (Line 161) and mitigate gradient conflicts (Line 90). As this is one contribution highlighted by the authors, I believe some ablation analysis or discussion on not using the 3 subspaces design should be included.
* In figure 1 and in Line 251, how do you quantify "best trade-off"? What is the metric used to determine this?
* Following on the previous question, in figure 1, since MNLI performance is the criteria and also the y-axis, why is the selected models (red) not always the best performing models?

Minor Questions:
* How large is the validation set used in Task-agnostic search (Line 206)? It is quite surprising to me that evaluating the 256 candidate subnetworks on this set (or on MNLI) takes less than one hour.
* Line 300. Not clear about the MLM baseline. Does it mean the SuperLM is directly trained on MLM objective? Is the "random sample a subnetwork" step still used?

**Limitations:**

Limitations are discussed in Appendix E.

**Strengths And Weaknesses:**

Strength:
* AutoDistill can be practically useful. It achieves 3x compression over other distillation methods. The resulting models can be re-used by practitioners and researchers.
* Solid experiment and clear visualization.
* The authors have uploaded code with running instructions to ensure reproducibility.

Weakness:
* The novelty of AutoDistill is still unclear. Components of AutoDistill seem to be adapted from existing methods.
* Writing is sometimes confusing and not coherent. See questions below.

---

> ### Author Response · Authors · 2022-08-02
> **Response to reviewer ov4n (Q1-Q4)**
>
> Thank you for the review and for bringing up these questions. Please see our responses below to clarify the main concerns.
>
> **Q1: The novelty of AutoDistil is still unclear. Components of AutoDistil seem to be adapted from existing methods.**
>
> A1: Please refer to the common response to all reviewers for our novelty and distinctions over prior work.
>
> **Q2: Does "few-shot" in the method mainly mean splitting the search space into K=3 subspaces?**
>
> A2: Yes, please refer to Table 1 for search space partition and lines 136-167 in paper for more details. We adopt the "few-shot" NAS terminology from prior work in the CV domain [1].
>
> [1] Zhao, Y., Wang, L., Tian, Y., Fonseca, R., \& Guo, T. (2021, July). Few-shot neural architecture search. In ICML.
>
> **Q3: What is the major difference between AutoDistil and Reference [16] (Zhao et al., 2021)?**
>
> A3: The main differences as also highlighted under **novelty** consist of the following:
>
> - **Fully task-agnostic SuperNet training**. {AutoDistil} training is fully task-agnostic in contrast to Zhao et al. [16] that uses task-specific NAS (see A1 (i-ii)). Task-agnostic NAS is challenging since we do not have access to task labels during training and we want to show generalization on evaluating diverse downstream NLU tasks in the GLUE benchmark. AutoDistil leverages self-attention distillation that is an unsupervised training objective. Incorporating self-attention loss for training and distillation in NAS is non-trivial as it requires aligning attention states of diverse student subnetworks and the large teacher model. We develop an extraction and alignment strategy (Section 3.2) to address this challenge.
>
> - **NLP vs. CV domain**. AutoDistil works on the NLP domain with the Transformer architecture (see Figure 2) under the pre-training and fine-tuning paradigm, while reference [16] works on the CV domain with a CNN architecture with different design and search spaces. Different from CV domain, NLP tasks have different objectives and evaluation metrics for classification (e.g., MNLI), regression (e.g., STS-B) and correlation (e.g., CoLA). Overall, the search space design (Section 3.1), SuperNet training with distillation and sub-network extraction strategy (Section 3.2) and search strategy (Section 3.3) are all quite different. While we briefly discuss these differences (lines 53-70), we will add a more elaborate discussion.
>
> **Q4: The motivation of few-shot NAS is to alleviate conflicts in weight-sharing (Line 161) and mitigate gradient conflicts (Line 90). As this is one contribution highlighted by the authors, I believe some ablation analysis or discussion on not using the 3 subspaces design should be included.**
>
> A4: We included this ablation analysis in Section 4.1.4 (lines 312-332). We compare the performance of a single space ($K=1$) corresponding to not using the 3 subspaces design against our few-shot design with multiple sub-spaces ($K=3$) with results in Table 5. We extract subnetworks with the same architecture ($6$ layers, $768$ hidden, $12$ heads, MLP ratio $4$) from trained SuperLMs for each strategy for evaluation with results in Table 5. We observe that our design strategy performs the best while containing lesser number of subnetworks demonstrating the benefit of few-shot NAS for language model distillation. We choose $K$=$3$ (i.e. 3 sub-spaces) for few-shot NAS for three reasons: (i) The 3 sub-spaces correspond to base, small and tiny model sizes. (ii) Searching over different values of $K$ is a very resource-extensive process since it requires training $K$ SuperLMs for each choice of $K$. (iii) As $K$ increases, the search process becomes similar to the undesirable brute-force discrete search that trains all models in search space individually.

---

> > ### Author Response · Authors · 2022-08-02
> > **Response to reviewer ov4n (Q5-Q8)**
> >
> > **Q5: In figure 1 and in Line 251, how do you quantify "best trade-off"? What is the metric used to determine this?**
> >
> > A5: We describe how to search for the optimal sub-network in Section 3.3 and Section 4.1. The "best trade-off" for optimal student selection is given by the sub-network with the least validation loss subject to the resource constraint as described in Eqn (6). For instance, we set a constraint in Eqn. (6) such that the #FLOPs of the optimal Base-sized task-agnostic compressed model is atleast 50\% less than the teacher model (lines 244-245). Since the SuperNet training is task-agnostic, the obtained student models have to be fine-tuned on downstream tasks to report the final task performance (similar to pre-train and fine-tune paradigm of BERT-like language models).
> >
> > **Q6: Following on the previous question, in figure 1, since MNLI performance is the criteria and also the y-axis, why is the selected models (red) not always the best performing models?**
> >
> > A6: Note that our objective is to minimize the \#FLOPs and maximize the accuracy (e.g., on MNLI) with the trade-off determined by the resource constraint (see A5) for different partitions (Base, Small, Tiny). Given a gallery of compressed models from AutoDistil with variable FLOPs and performance, we use A5 for optimal model selection.
> >
> > Another potential reason why red is always not the best model is that we use the heldout validation set from the unlabeled training corpus (Wikipedia + BooksCorpus) for student selection and then evaluate them on MNLI (see Section 3.3) which may not be optimal due to sample differences in the two datasets.
> >
> > **Q7: How large is the validation set used in Task-agnostic search (Line 206)? It is quite surprising to me that evaluating the 256 candidate subnetworks on this set (or on MNLI) takes less than one hour.**
> >
> > A7: Thanks for raising this point.
> > (i) The validation set contains $300K$ instances. We use $128$ as the sequence length and batch size. (ii) Table 3 reports the search cost for Small-sized models (line 282) from {AutoDistil} and AutoTinyBERT. Note that this step does not require any training for {AutoDistil}. We compute only the self-attention relation loss for all the $256$ student subnetworks ($5x-22x$ speedups for Small-sized models) using Equation 4 with the teacher relations computed only once. We use Equation 6 to select the subnetwork with desired trade-off with deterministic computation of the FLOPs. The algebraic expression to compute FLOPs as a function of layers, heads, hidden size etc. is included in the submitted source code. In contrast, AutoTinyBERT performs task-specific search which requires fine-tuning the subnetworks on the task (e.g., MNLI) thereby increasing the search cost.
> >
> > **Q8: Line 300. Not clear about the MLM baseline. Does it mean the SuperLM is directly trained on MLM objective? Is the "random sample a subnetwork" step still used?**
> >
> > A8: "MLM" indicates that the SuperLM is trained with masked language modeling (MLM) loss instead of using Equation 4 for self-attention distillation loss. The remaining steps including the random sampling of subnetworks are the same. Please refer to Section 4.1.3 and Table 4 for the result comparison.

---

> ### Author Response · Authors · 2022-08-08
> **Requesting Acknowledgment of Our Response**
>
> Dear Reviewer,
>
> We have incorporated your suggested clarifications and our prior responses in the revised version (changes highlighted in blue color). Due to limited space in the main manuscript, most of the discussions are added to the Appendix in the submitted Supplementary.
>
> Please consider revising your score if our clarifications alleviate your concerns, or let us know if you have any additional questions.

---

### Official Review · Reviewer_1T87 · 2022-07-12

**Rating:** 7
**Confidence:** 4
**Soundness:** 4 excellent
**Presentation:** 3 good
**Contribution:** 3 good

**Summary:**

This work partitions the search space of neural architecture search into K sub-spaces, trains a SuperLM for each partition with task-agnostic self-attention distillation, and finds optimal students without additional training.

**Questions:**

How you define the search space in Table 1 is not straightforward, though the authors mentioned it is based on inductive bias and heuristic. Could you elaborate on the heuristic part? The prior work mentioned in Line 129 is not evaluated by Transformer architectures, so I doubt inductive bias from this work is valid to apply.

SuperLM in a smaller region is also initialized with pre-trained BERT-Base, so only the leftmost layers, dimensions, and heads are used. However, dimensions and heads are order-independent, meaning that the performance might change after re-ordering.

Section 4.1.3 and Section 4.1.4 are only tested on MRPC and RTE datasets. I am suspicious they are enough or good representatives.

AutoDistil-proxy only uses MNLI as a source task. I am curious how the transferability of optimal architectures changes between source and target tasks?

**Limitations:**

I wonder whether the few-shot task-agnostic NAS method is applicable to other domains, including computer vision.


**Strengths And Weaknesses:**

The proposed methods outperform traditional knowledge distillation baseline and other NAS baselines. The idea of partitioning search space by using multiple supernets is intuitive and practically useful. Other implementations are borrowed from the existing methods like BigNAS and MiniLM. The paper provides adequate ablation study results to confirm the effectiveness of each component (e.g., training strategies, few-shot NAS, search strategy).

---

> ### Author Response · Authors · 2022-08-02
> **Response to reviewer 1T87 (Q1-Q2)**
>
> Thank you for the valuable comments. Please find our responses as follows.
>
> **Q1: Could you elaborate on the heuristic part of the search space design? The prior work mentioned in Line 129 is not evaluated by Transformer architectures, so I doubt inductive bias from this work is valid to apply.**
>
> A1: The details of the search space is summarized in Table 1. In general, we partition the whole Transformer search space into $K=3$ sub-spaces such that each sub-space covers different sizes of student models (by number of parameters) depicting Tiny, Small and Base model sizes. Given a BERT-sized teacher model (109$M$ params), we roughly set the partition thresholds for Tiny, Small and Base sizes at 10$M$, 40$M$ and 80$M$ params. From Table 2 (\#Para column), we observe that each partition contains compressed models from prior work -- allowing us to fairly compare the models in each partition on accuracy vs. params/FLOPs.
>
> For our search space, each partition still contains thousands of candidate subnetworks not all of which are useful. Now, we leverage two primary heuristics (lines 129-136): (i) we constrain all layers in a sampled student subnetwork to be homogeneous i.e., the same number of attention heads, hidden dimension, etc. This not only reduces the search space, it is also more friendly to hardware and software frameworks. (ii) Motivated by previous work [1-2] showing that thinner and deeper neural networks have better representation capabilities and perform better than wider and shallower neural networks, we designed sub-spaces with deeper layers (e.g., $4-7$ for Tiny, $9-12$ for Small and Base) and compute the range of hidden dimensions to meet the overall model parameter budget in each partition. Additional constraints arise from Transformer design principles, for instance, hidden size is always a multiple of the number of attention heads. While the above steps require enumeration of different subnetwork architectures, this is typically fast given an algebraic expression to compute model parameters as a function of layers, heads, hidden size, etc. (included in submitted source code), does not require any training, and a one-time process depending only on the teacher model architecture. We will add this discussion in paper.
>
> Thanks to the reviewer for pointing out the different convolution architecture in prior cited work [1]. A more relevant reference for the Transformer architecture is given in [2]. Please refer to Figure 2(b) in [2] that shows the impact of Transformer depth on MNLI accuracy given similar overall model parameters.
>
> [1] Romero, A., Ballas, N., Kahou, S. E., Chassang, A., Gatta, C., \& Bengio, Y. (2015). Fitnets: Hints for thin deep nets. In ICLR.
>
> [2] Li, Z., Wallace, E., Shen, S., Lin, K., Keutzer, K., Klein, D., \& Gonzalez, J. (2020). Train big, then compress: Rethinking model size for efficient training and inference of transformers. In ICML.
>
> **Q2: SuperLM in a smaller region is also initialized with pre-trained BERT-Base, so only the leftmost layers, dimensions, and heads are used. However, dimensions and heads are order-independent, meaning that the performance might change after re-ordering.**
>
> A2: In order to extract student sub-networks from SuperLM (line 163), AutoDistil uses a simple truncation strategy, i.e., bottom-left extraction. This is used during sampling sub-networks for SuperLM training via weight sharing after which the **sub-network weights are updated** via self-attention relation distillation. There can be better strategies to extract sub-networks (e.g., ordering heads by importance) that we defer to future work. We did explore some strategies for selecting layers (alternate vs. top vs. bottom) with results in Section B of Appendix.

---

> > ### Author Response · Authors · 2022-08-02
> > **Response to reviewer 1T87 (Q3-Q5)**
> >
> > **Q3: Section 4.1.3 and Section 4.1.4 are only tested on MRPC and RTE datasets. I am suspicious they are enough or good representatives.**
> >
> > A3: MRPC and RTE represent paraphrase (i.e. semantic textual similarity) detection and natural language inference tasks, respectively. These tasks are low-resource tasks in the GLUE benchmark [1] since they have limited number of training examples (~$3K$ labels for each) for model fine-tuning. This allows us to evaluate the transferability of the compressed models trained on unlabeled general domain corpus (e.g., Wikipedia, BooksCorpus) and fine-tuned on the above downstream tasks with limited labeled data.
> >
> > We conducted additional experiments on MNLI; where we have a large dataset containing $400K$ labeled examples. Prior work [2] has shown MNLI to transfer well to other tasks in the GLUE benchmark. We observe the following performance of different task-agnostic training strategies (Table 4) on the MNLI task: 82.2 (MLM), 83.5 (KD_{att}+Cont.), and 83.2 (KD$_{att}$), respectively. As with MRPC and RTE, we make similar observations. (i) Self-attention distillation strategy performs better than MLM for SuperLM training. (ii) There is limited performance gain with continued training (thereby increased computation cost) as done in existing works that demonstrate the effectiveness of our single-stage training protocol. We will add these discussions in paper.
> >
> > [1] Wang, A., Singh, A., Michael, J., Hill, F., Levy, O., \& Bowman, S. R. (2018). GLUE: A multi-task benchmark and analysis platform for natural language understanding. arXiv preprint arXiv:1804.07461.
> >
> > [2] Chen, T., Frankle, J., Chang, S., Liu, S., Zhang, Y., Wang, Z., \& Carbin, M. (2020). The lottery ticket hypothesis for pre-trained bert networks. In NeurIPS.
> >
> > **Q4: AutoDistil-proxy only uses MNLI as a source task. I am curious how the transferability of optimal architectures changes between source and target tasks?**
> >
> > A4: This is a very interesting question. This has been studied with regards to model pruning in the lottery ticket hypothesis work [1] for BERT. They observe that transferability seems to correlate with the number of training examples available for the source task. This is particularly beneficial with MNLI containing a large number of training examples as compared to other low-resource tasks in the GLUE benchmark. Similar to [1], we also observe MNLI to transfer well to other tasks in the GLUE benchmark with AutoDistil-proxy even outperforming task-specific NAS methods like DynaBERT (Table 2) on both parameters (26.1$M$ vs. 37.7$M$) and average accuracy (79.9 vs. 77.4).
> >
> > [1] Chen, T., Frankle, J., Chang, S., Liu, S., Zhang, Y., Wang, Z., \& Carbin, M. (2020). The lottery ticket hypothesis for pre-trained bert networks. In NeurIPS.
> >
> > **Q5: I wonder whether the few-shot task-agnostic NAS method is applicable to other domains, including computer vision.**
> >
> > A5: Most NAS works in computer vision (CV) (e.g., Once-for-all, One-Shot NAS) leverage hard class labels from a given task (e.g., image classification). They often use similar training recipes for SuperNets as in ImageNet-trained models (e.g., MobileNet, RegNet) for task-specific optimization with accuracy as an evaluation metric. In contrast, the few-shot task-agnostic NAS strategy used in AutoDistil training is fully task-agnostic and does not access task labels during SuperNet training.
> >
> > A potential method to adopt this strategy for the CV domain is to consider a self-supervised learning framework like SimCLR [1] that leverages data augmentation for consistency learning. This requires both a self-supervised teacher like SimCLR and a self-supervised training objective (e.g., self-attention relation distillation for Transformers or architecture-agnostic consistency learning). This would be a very interesting direction for future work.
> >
> > [1] Chen, T., Kornblith, S., Norouzi, M., \& Hinton, G. (2020). A simple framework for contrastive learning of visual representations. In ICML.

---

### Official Review · Reviewer_swEZ · 2022-07-16

**Rating:** 7
**Confidence:** 4
**Soundness:** 4 excellent
**Presentation:** 3 good
**Contribution:** 3 good

**Summary:**

This paper proposes an effective method AutoDistill to compress several student models with various computation costs at once. By combining few-shot NAS and task-agnostic knowledge distillation, AutoDistill alleviates optimization interference and speed-up the training process, enabling single-stage training without further pre-training and distillation. Experimental results demonstrate that AutoDistill outperforms strong NAS and knowledge distillation baselines.

**Questions:**

1. How transferable is the optimal student architecture derived from the proposed AutoDistill method across teachers and knowledge distillation objectives?
2. Given some optimal student architecture derived by NAS, how would traditional task-agnostic knowledge distillation method (e.g., MiniLMv2) perform?

**Limitations:**

The authors address the limitations adequately.

**Strengths And Weaknesses:**

Strengths:
1. The paper is clearly written and well structured. The authors first demonstrate the weaknesses of existing knowledge distillation and NAS-based language model compression methods, then they introduce their method clearly, making it easy for readers to follow.
2. The proposed method is reasonable and technically sound. By adopting the idea of few-shot NAS, the problem of optimization interference is alleviated. Moreover, by fully task-agnostic knowledge distillation and search, AutoDistill significantly reduces training costs.
3. The authors conduct comprehensive experiments and the results verify the superiority of AutoDistill over strong baseline methods.

Weaknesses:
1. Although the proposed method is effective, the major two components of the method (i.e., few-shot NAS and task-agnostic knowledge distillation) are not novel. When compared with DynaBERT, it seems that AutoDistill is its task-agnostic version with a larger searching space and a few other modifications.
2. When compared with existing NAS-based language model compression method (e.g., AutoTinyBERT), AutoDistill enjoys much less computation cost in training and searching. However, when compared with traditional task-agnostic knowledge distillation method (e.g., MiniLMv2), the advantage of training speed is gone.

---

> ### Author Response · Authors · 2022-08-02
> **Response to reviewer swEZ (Q1-Q2)**
>
> We thank the reviewer for insightful comments and for acknowledging the value of our work. Please find our responses to the questions as follows.
>
> **Q1: Although the proposed method is effective, the major two components (i.e., few-shot NAS and task-agnostic knowledge distillation) are not novel. When compared with DynaBERT, it seems that AutoDistill is its task-agnostic version with a larger searching space and a few other modifications.**
>
> A1: Prior work on few-shot NAS to obtain multiple compressed models of varying FLOPs are task-specific and developed for computer vision (CV) domain. Traditional works on task-agnostic knowledge distillation target a specific compressed model architecture. However, it is non-trivial to obtain a combination of the above. For instance, task-agnostic self-attention distillation for SuperNet training and distillation with NAS requires aligning the attention states (query, key, value) of varying size subnetworks to that of the large teacher. We develop an extraction and alignment strategy (Section 3.2) to address this challenge. During sampling, we employ Sandwich rule (lines 173-176) to improve the performance of all subnetworks by increasing the performance lower bound (smallest subnetwork) and upper bound (largest one) across all subnetworks.
>
> Further, existing NAS works in NLP (e.g., DynaBERT, NASBERT, AutoTinyBERT) use additional expensive step(s) of further pre-training / distillation of the optimal architecture with task labels for best performance. In contrast, our single-stage task-agnostic method without additional training offers massive reduction in computational cost for training and search (see Table 3).
>
> Compared with DynaBERT, (i) our search space is more fine-grained. For instance, we independently search for width, depth, heads, MLP ratio etc. as opposed to searching for a constant depth ($m_d$) or width multiplier ($m_w$) in DynaBERT which only considers $12$ possible combinations of $m_d$ and $m_w$; (ii)  our training objective does not require labels and is fully task-agnostic with subnetwork attention state alignment for self-attention relation distillation; (iii) further, AutoDistil uses few-shot NAS (Table 1) to mitigate gradient conflicts in SuperNet training, while DynaBERT applies one-shot NAS; (iv) DynaBERT uses additional tricks like data augmentation and teacher assistant also specific to each task, whereas AutoDistil uses a single-stage task-agnostic training resulting in reduced computational cost.
>
> Table 1 in the Appendix compares AutoDistil against all recent NLP works on NAS (e.g., DynaBERT, AutoTinyBERT, NASBERT) and Distillation (MiniLM, DistilBERT, MobileBERT, PKD). Key experimental comparisons are summarized in Figure 1 and Table 2 of the main paper and Figure 1 of the Appendix. We will add these discussions to our revision.
>
> **Q2: When compared with existing NAS-based language model compression method (e.g., AutoTinyBERT), AutoDistill enjoys much less computation cost in training and searching. However, when compared with traditional task-agnostic knowledge distillation method (e.g., MiniLMv2), the advantage of training speed is gone.**
>
> A2: Different hardwares (e.g., FPGA, CPU, GPU) have different resource constraints. AutoDistil generates a gallery of fully trained compressed student models with variable resource constraints (e.g., FLOPs, parameters) using NAS. One can simply choose a model from the trained pool given the resource constraint and only fine-tune on the downstream task. In contrast, traditional task-agnostic knowledge distillation (KD) methods (e.g., MiniLM) target specific compression rate and needs to be trained repeatedly for different student configurations (corresponding to different resource constraints). Therefore, AutoDistil has a much reduced amortized computation cost even considering traditional KD methods. Further, traditional methods require several trial and errors to come up with a viable candidate architecture given a constraint before running the KD algorithm.

---

> > ### Author Response · Authors · 2022-08-02
> > **Response to reviewer swEZ (Q3-Q4)**
> >
> > **Q3: How transferable is the optimal student architecture derived from the proposed AutoDistill method across teachers and knowledge distillation objectives?**
> >
> > A3: This is a great question. In general, a better teacher model leads to a better student model [1-2] during distillation. We adopted BERT as a teacher for a fair comparison with existing works with the same teacher. We also compare with different training objectives like self-attention distillation and masked language modeling (Table 4) and demonstrate the former to work better for our SuperNet training. We demonstrate transferability by training the {AutoDistil} students in a task-agnostic manner and evaluating on different downstream tasks (Table 2). Note that these tasks are quite diverse ranging from classification (e.g., MNLI), regression (e.g., STS-B) and correlation (e.g., CoLA). We also demonstrate this to work better or comparable to task-specific NAS methods (e.g., DynaBERT, AutoTinyBERT) with further reduction of computational cost.
> >
> > [1] Gou, J., Yu, B., Maybank, S. J., \& Tao, D. (2021). Knowledge distillation: A survey. International Journal of Computer Vision, 129(6), 1789-1819.
> >
> > [2] Wang, W., Wei, F., Dong, L., Bao, H., Yang, N., \& Zhou, M. (2020). Minilm: Deep self-attention distillation for task-agnostic compression of pre-trained transformers. In NeurIPS.
> >
> > **Q4: Given some optimal student architecture derived by NAS, how would the traditional task-agnostic knowledge distillation method (e.g., MiniLMv2) perform?**
> >
> > A4: In practice, additional or continued training of the optimal student architecture has demonstrated increased task performance with increased computational cost as in AutoTinyBERT (cost comparison in Table 3). The major advantage of AutoDistil is a single stage training scheme without additional training. We do perform an ablation in Table 4 where we continue training the searched model with self-attention distillation for additional steps referred as `KD$_\text{att}$+Cont.' (similar to MiniLM). But we did not observe any significant gains on a subset of the tasks.

---

### Author Response · Authors · 2022-08-02
**Common Response to all Reviewers (ii)**

We would also like to summarize our novelty and distinctions over prior work.

**(i) Task-agnostic (AutoDistil) vs. Task-specific NAS.**
NAS works in computer vision (CV) (e.g., Once-for-all, One-Shot NAS) leverage hard class labels from a given task (e.g., image classification). They often use similar training recipes for SuperNets as in ImageNet-trained models (e.g., MobileNet, RegNet) for task-specific optimization with accuracy as an evaluation metric.
In contrast, **AutoDistil training is fully task-agnostic** and does not access task labels during SuperNet training. Different from CV domain, NLP tasks have different objectives and evaluation metrics for classification (e.g., MNLI), regression (e.g., STS-B) and correlation (e.g., CoLA).

**(ii) Fully task-agnostic training objective.**
In view of the above, our SuperNet training objective leverages self-attention distillation which is unsupervised and does not require task labels or additional training. In contrast, for downstream task adaptation, existing NAS works in NLP (e.g., DynaBERT, NASBERT, AutoTinyBERT) use additional expensive step(s) of further pre-training / distillation of the optimal architecture with task labels for best performance.
Incorporating self-attention loss for SuperNet training and distillation in NAS is non-trivial. It requires aligning the attention states (query, key, value) of varying size subnetworks to that of the large teacher. We develop an extraction and alignment strategy (Section 3.2) to address this challenge. During sampling, we employ Sandwich rule (lines 173-176) to improve the performance of all subnetworks by increasing the performance lower bound (smallest subnetwork) and upper bound (largest one) across all subnetworks.

**(iii) Single-stage training for computational savings.**
In contrast to prior works, we do a single-stage training combining NAS and distillation with no further pre-training or augmentation and demonstrate the superior performance of the NAS process itself. Obtained subnetworks are simply fine-tuned on downstream tasks. Table 3 demonstrates a massive reduction in search and additional training cost over state-of-the-art NAS work (AutoTinyBERT) on NLP tasks.

**(iv) One-shot vs. Few-shot NAS.**
In contrast to prior NAS works in the NLP domain (e.g., DynaBERT, AuotTinyBERT, NASBERT) that employ a single large search space (One-shot NAS), we demonstrate the value of sub-space partitioning to reduce gradient conflicts and optimization interference for improved performance with Few-shot NAS design and ablation analysis.

---

### Author Response · Authors · 2022-08-02
**Common Response to all Reviewers (i)**

We thank all reviewers for their insightful comments. The main contributions of this work as stated by the reviewers are as follows:

- Methods for sub-space partitioning for alleviating optimization inference and task-agnostic distillation are reasonable and technically sound **[Reviewers swEZ, 1T87]**
- Reduced training cost and computationally efficient method **[Reviewers swEZ, 1T87, ov4n]**
- Solid experiments and clear visualization; ablation of each component (e.g., training strategies, few-shot NAS, search strategy) **[Reviewers swEZ, 1T87, ov4n]**
- Superiority over strong NAS and KD baselines **[Reviewers swEZ, 1T87, ov4n]**
- The paper is clearly written, well structured and easy to follow **[Reviewer swEZ]**
- Reproducible (submitted code with running instructions); resulting model checkpoints can be re-used by practitioners and researchers **[Reviewer ov4n]**

Our strengths over existing NLP and CV works:

1. Single-stage training (no further pre-train / augment / KD).
2. Fully task-agnostic training with subnetwork attention state alignment for self-attention relation distillation.
3. Few-shot NAS to mitigate gradient conflicts in SuperNet training vs. One-shot NAS (e.g., AutoTinyBERT, NASBERT, DynaBERT).
4. Strong results over all NAS and distillation works in NLP (Figure 1) with $2.7x$ additional compression over best performing compression technique in literature with upto $41x$ reduction in FLOPs over large teacher corresponding to  AutoDistil-Tiny with some loss in task performance.

---

### Author Response · Authors · 2022-08-09
**Revision Summary**

We submitted a revised manuscript with the following changes (highlighted in blue color). Most of these are added to the Appendix in Supplementary due to limited space in main manuscript.

(1) Additional discussion on our differences with existing works on NAS and KD with regards to our fine-grained search space, different training and search strategies (Section A of Appendix).

(1.1) These include DynaBERT (task-specific NAS with shallow search space in Section A.3), Few-shot NAS in Computer Vision (task-specific NAS, different domain with CNN architecture and search space in Section A.2) in contrast to our task-agnostic NAS with fine-grained Transformer search space.

(1.2) Discussion on amortized training cost of using traditional KD methods (e.g., MiniLM) with pre-specified compression rate  (Section A.1) and representative experiment ('KD_attn+Contd.' in Table 4 of main paper).

Note that we have already compared against these works in the NLU domain as baselines on the GLUE benchmark in our main manuscript (Table 2) demonstrating better trade-off in overall task performance vs. computational cost (e.g., #Params, #FLOPs).

(2) Discussion on transferability of optimal subnetworks (Section E of Appendix), additional experiments on MNLI (Section 4.1.3 and Table 4 in main paper) for ablation, reference of prior work on Lottery Ticket Hypothesis for BERT (Chen et al., NeurIPS 2020) demonstrating MNLI to be a good transfer task.

(3) Additional discussion on search space design (Section D of Appendix), inductive bias with reference of prior work (Li et al., ICML 2020) studying impact of network depth on Transformer performance

(4) Additional discussion on optimality of subnetworks and the best trade-off in task performance vs. computational cost (Sections G, H, I) in Appendix.

(5) Applicability of task-agnostic NAS for other domains like Computer Vision (Section F of Appendix).

---

### Meta-Review · Area_Chair_u3YT · 2022-08-29

**Recommendation:** Accept
**Confidence:** Less certain

**Metareview:**

The submission introduces an approach to searching for student architectures to distill large language models into. The authors divide the search space into different sizes of student model, train a SuperLM for each model, and then select the optimal student from within these using knowledge distillation. While several of the techniques here have been used before, the reviewers found that the overall approach was well motivated and effective empirically, outperforming strong baselines. As pointed out by reviewer ov4n, the writing could be improved to make the paper more accessible to people less familiar with NAS, but overall this is solid work and I recommend acceptance.

**Award:**

No

---

### Decision · Program_Chairs · 2022-09-14

Accept